# Study on Roasting for Selective Lithium Leaching of Cathode Active Materials from Spent Lithium-Ion Batteries

Yeonjae Jung [1,2], Bongyoung Yoo [1], Sungcheol Park [2], Yonghwan Kim [3] and Seongho Son [2,*]

[1] Department of Materials Science and Chemical Engineering, Hanyang University, Ansan 15588, Korea; yeonjae89@kitech.re.kr (Y.J.); byyoo@hanyang.ac.kr (B.Y.)

[2] Heat & Surface Technology R&D Department, Korea Institute of Industrial Technology, Incheon 21999, Korea; schpark@kitech.re.kr

[3] Industrial Materials Processing R&D Department, Korea Institute of Industrial Technology, Incheon 21999, Korea; yhkim@kitech.re.kr

\* Correspondence: shson@kitech.re.kr; Tel.: +82-032-850-0242

**Abstract:** Recently, many studies have been conducted on the materialization of spent batteries. In conventional cases, lithium is recovered from an acidic solution through the leaching and separation of valuable metals; however, it is difficult to remove impurities because lithium is recovered in the last step. Cathode active materials of lithium-ion batteries comprise oxides with lithium, such as $LiNi_xCo_yMn_zO_2$ and $LiCoO_2$. Thus, lithium should be converted into a compound that can be leached in deionized water for selective lithium leaching. Recent studies on the leaching and recovery of $Li_2CO_3$ through a carbon reduction reaction show low economic efficiency, due to the solubility of $Li_2CO_3$ at room temperature being as low as 13 g/L. This paper proposes a method of roasting after nitric acid deposition for selective lithium leaching and recovery to $LiNO_3$. Based on experiments involving the varying of the amount of nitric acid, roasting temperature, and solid–liquid ratio, optimal values were found to be dipping in 10 M $HNO_3$ 2 mL/g, roasting at 275 °C, and deionized water with a solid–liquid ratio of 10 mL/g, respectively. Over 80% Li leaching was possible under these conditions. IC analysis confirmed that the purity was 97% lithium nitrate.

**Keywords:** cathode active materials; selective lithium leaching; roasting

## 1. Introduction

In recent years, the use of secondary batteries has increased rapidly owing to the increasing use of mobile devices and electric vehicles. In particular, there has been a significant increase in their usage in the case of electric vehicles, where large-capacity batteries are used, unlike those used in mobile devices. According to the International Energy Agency, electric vehicle usage is expected to grow rapidly, reaching 220 million vehicles by the 2030s [1–3]. For electric vehicles and mobile devices, lithium-ion batteries (LIBs) with high energy density and low weight characteristics are commonly used [4,5]. Therefore, it is important to investigate the treatment and recycling technology of spent LIBs, which are increasing in quantity with the increased market size of batteries.

Cathode active materials in LIBs contain approximately 4–7% lithium (Li). It is difficult and costly to remove impurities using conventional processes, because Li recovery is conducted in the last step. To overcome this problem, selective Li leaching has been actively studied [6–8], and involves leaching only Li after changing the phase of Li into a leachable phase by performing a carbon reduction reaction. In this reaction, a portion of Li in $LiCoO_2$ in the active material reacts with carbon (C), thereby forming $Li_2CO_3$.

$$4LiCoO_2 + 3C \rightarrow 2Li_2CO_3 + 4Co + CO_2(g) \tag{1}$$

If an organic material containing C exists, $LiNi_xCo_yMn_zO_2$ can be reduced to $Li_2CO_3$, Ni, Co, and MnO through a carbon reduction reaction. After the reduction heat treatment,

only $Li_2CO_3$ could be selectively leached when leaching was performed in deionized (DI) water. However, in the case of $Li_2CO_3$, the solubility at 25 °C is very low at 12.9 g/L. Therefore, the solid–liquid ratio of the leaching should be large. On the other hand, $LiNO_3$ has a very high solubility of 900 g/L [9]. Therefore, for efficient selective lithium leaching and recovery, leaching is required after conversion to $LiNO_3$. This study was conducted on roasting to leach and recover the highly soluble $LiNO_3$.

## 2. Experimental

### 2.1. Materials

Spent LIB powder (black powder) used in the experiments was acquired from a battery recycling company in South Korea. The powder was crushed and ground after physio-chemical separation. The metal content in the spent secondary battery powder varied depending on the characteristics. X-ray diffraction (XRD, X'Pert-pro mPD, PANalytical, Malvern, UK) was performed to confirm the composition of the powder, and the proportions of different metals were analyzed by inductively coupled plasma optical emission spectroscopy (ICP-OES, Integra XL, GBC Scientific, Braeside, Australia) after leaching in aqua regia.

As shown in Figure 1, the XRD results show that there are carbon and oxides containing lithium, such as NCM. The black powder was dissolved in aqua regia solution to determine its chemical composition using ICP-OES. Detailed results are presented in Table 1. The main metals identified in the powder were Co, Ni, Mn, and Li, along with trace amounts of Al, Fe, and Cu. All other reagents used in this work were of analytical grade, and all solutions were prepared using distilled water.

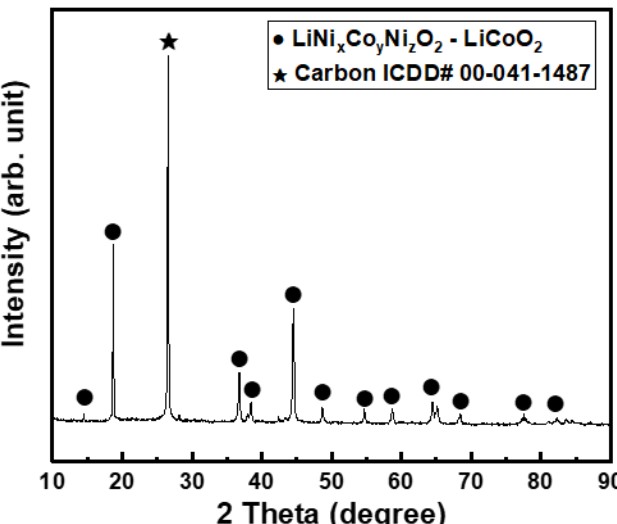

**Figure 1.** Spent LIB powder XRD result.

**Table 1.** Spent LIB powder ICP analysis result.

| ICP Analysis | Component (Wt. %) | | | | | | |
|---|---|---|---|---|---|---|---|
| | Ni | Co | Mn | Li | Al | Cu | Fe |
| Black powder | 15.63 | 8.37 | 7.64 | 3.92 | 3.20 | 1.15 | 0.38 |

### 2.2. Experimental Procedure

In this study, selective Li leaching affects the evaluation of pretreatment, and roasting experiments were conducted using the sequence shown in Figure 2. The pretreatment was conducted at varying temperatures (550–800 °C) and time durations (0–40 h) to remove organic compounds, including carbon. The nitric acid leaching was performed at varying

solid–liquid ratios (0.2–25 mL/g) of 10 M nitric acid. For roasting to decompose nitrate compounds, excluding lithium nitrate, the temperature (200–700 °C) and time (1–10 h) were varied. After roasting, leaching was performed with DI water at room temperature at varying solid–liquid ratios (1–30 mL/g).

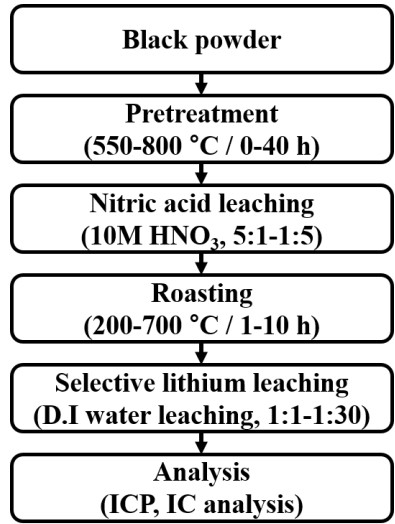

**Figure 2.** Experimental schematic diagram of selective lithium leaching through nitric acid leaching and roasting.

### 2.2.1. Pretreatment

The carbon contained in the black powder remained on the powder surface after nitrate leaching and roasting, thereby hindering selective Li leaching. Therefore, carbon should be removed through pretreatment. In this study, we performed pretreatment for 10 h at 550–800 °C, and compared the results of selective Li leaching in DI water after nitric acid leaching and roasting. For nitric acid leaching, 2 mL/g of 10 M nitric acid was used, and the roasting was performed for 10 h at 275 °C.

### 2.2.2. Nitric Acid Leaching

Pretreated black powder exists as lithium compounds, such as $LiCoO_2$ and $LiNi_xCo_yMn_zO_2$. A nitric acid leaching process is required for the conversion of nitric compounds such as $Ni(NO_3)_2$, $Co(NO_3)_2$, and $Mn(NO_3)_2$. The effect of the solid–liquid ratio of nitric acid on selective lithium leaching was investigated. The leaching solid–liquid ratio was carried out with 10 M nitric acid at 0.2–25 mL/g. Leaching was carried out in an alumina crucible, without stirring. After leaching for 10 min, roasting was performed at 275 °C for 10 h, and the lithium leaching ratio was analyzed after leaching in DI water at a solid–liquid ratio of 10 mL/g.

### 2.2.3. Roasting

For selective lithium leaching, it is necessary to convert compounds other than lithium nitrate. In this study, roasting was performed for 10 h at roasting temperatures ranging from 200 to 700 °C, with the optimum roasting temperature being 275 °C, and the experiment was conducted to determine the optimum roasting time by varying the time from 1 to 10 h. This experiment involved nitric acid leaching with 2 mL/g of 10 M nitric acid, and after roasting, the leaching rate was analyzed after leaching with 10 mL/g of DI water.

### 2.2.4. Selective Lithium Leaching in DI Water

After nitric acid leaching and roasting, leaching was performed in DI water. At room temperature, the solid–liquid ratio was changed from 5:1 to 1:5 to perform selective lithium leaching and then solid–liquid separation. The solution was analyzed for the lithium

leaching ratio by performing ICP analysis. In this study, the leaching ratio of $\eta_i$. could be calculated based on Equation (2):

$$\eta_i = \frac{C_i V}{M_0 W_0} \times 100\% \qquad (2)$$

where $C_i$ (gL$^{-1}$) and $V$ (L) are the concentration of element i and volume of leaching solution, respectively, and $M_0$ (g) and $W_0$ are the mass of the cathode active powder and the weight content of element $i$, respectively.

The liquid obtained through solid–liquid separation was dried at 90 °C for 48 h to obtain powder. XRD of the powder was then performed, with the results confirming the recovery of LiNO$_3$. In addition, the purity of LiNO$_3$ was analyzed based on the results of the IC analysis.

## 3. Results and Discussion

### 3.1. Effect of Pretreatment

According to a previous report, the decomposition of organic matter in lithium batteries begins at 350 °C [10]. Thermogravimetry differential scanning calorimetry (TG-DSC) analysis was performed to determine the temperature for the removal of organic substances, including carbon, in the black powder. The analysis was conducted at a temperature increase rate of 5 °C/min at atmospheric conditions. As shown in Figure 3, organic matter containing carbon was removed at temperatures above 437 °C. However, temperatures above 500 °C were also suitable for effective removal. A weight loss of approximately 44.76% was observed as the temperature increased from room temperature to 800 °C. There were two exothermic peaks in the DSC curve, appearing at approximately 500 and 700 °C.

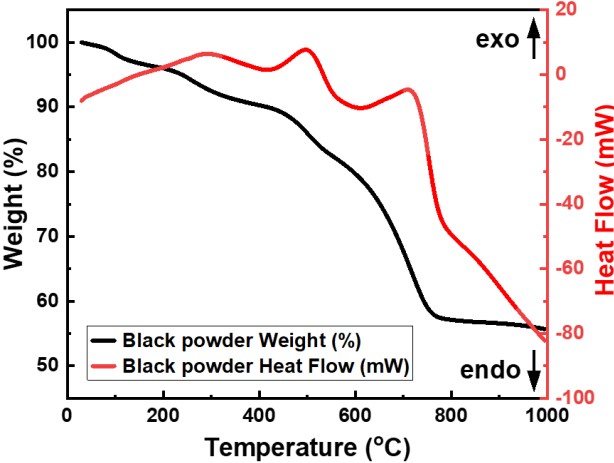

**Figure 3.** Thermogravimetric analysis in air atmosphere.

Nitric acid leaching and roasting were performed after pretreatment at 550–800 °C. The analysis results for lithium leaching in DI water are shown in Figure 4. The lithium leaching ratio was the highest when pretreatment was performed at 600 °C. When the time was varied at the optimal pretreatment temperature, most organic materials could be removed within 5 h, and the Li leaching ratio was high, as shown in Figure 4b. The results of the XRD analysis according to pretreatment are shown in Figure 5. It can be seen that the removal of organic matter, including carbon, through pretreatment improves selective lithium leaching, and we can conclude that for the removal of carbon-containing organic materials, pretreatment should be performed for over 5 h at 600 °C.

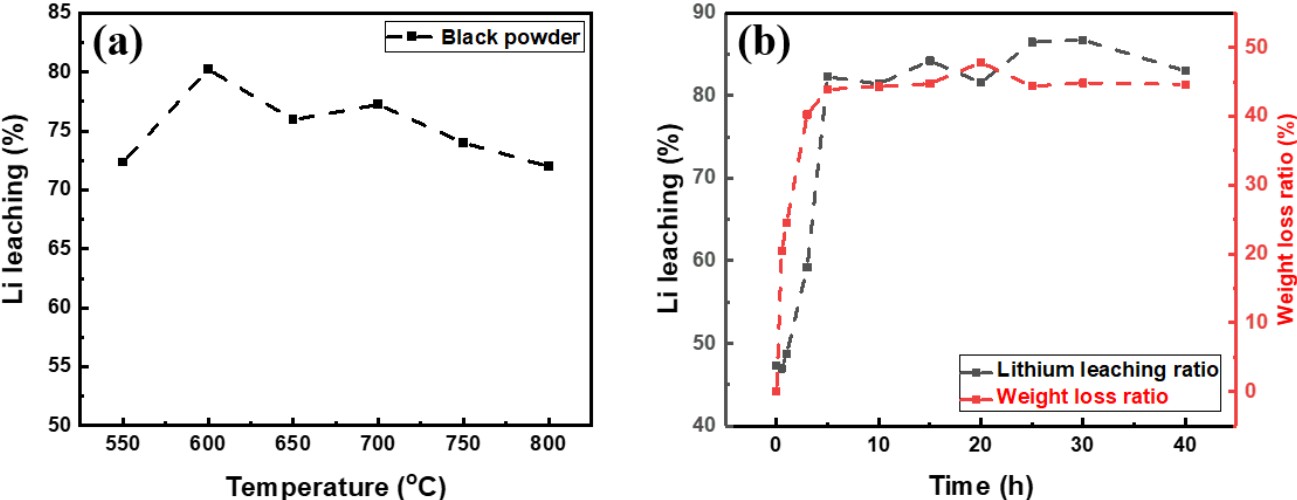

**Figure 4.** Variation in (**a**) lithium leaching ratio with pretreatment temperature, (**b**) weight loss and lithium leaching ratio with time at pretreatment temperature of 600 °C.

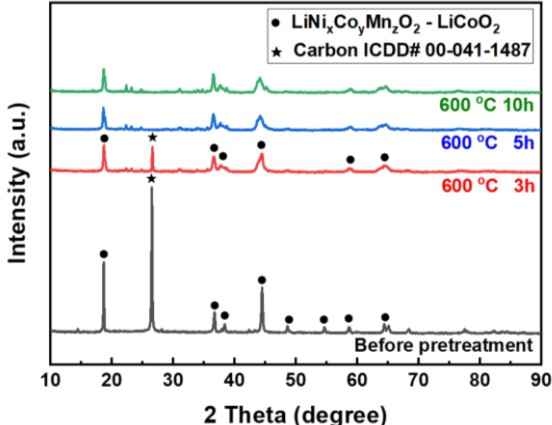

**Figure 5.** XRD according to pretreatment time at 600 °C.

### 3.2. Effect of Nitric Acid Leaching

Nitric acid leaching was performed using 10M nitric acid with pretreated powder, and $LiNi_xCo_yMn_zO_2$ and $LiCoO_2$ were present as nitrate compounds of $LiNO_3$, $Co(NO_3)_3$, $Ni(NO_3)_2$, and $Mn(NO_3)_2$. The results of selective lithium leaching with different amounts of 10 M nitric acid are shown in Figure 6. The lithium leaching ratio did not increase for 10 M nitric acid leaching of over 1 mL/g. This was removed at a temperature of 87.8 °C or higher when roasting residual nitrate that did not react with the black powder [11]. For this reason, nitric acid concentrations above 1 mL/g do not affect the selective lithium leaching.

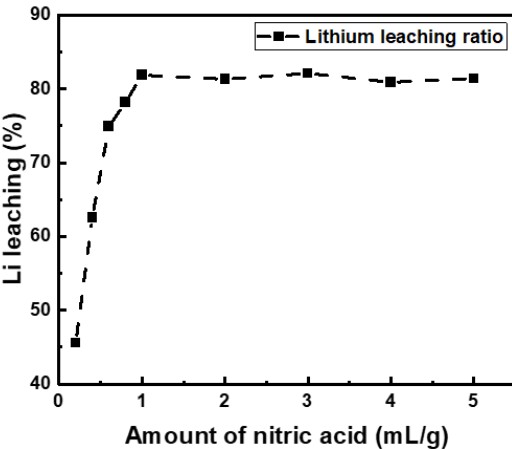

**Figure 6.** Variation in lithium leaching ratio with amount of nitric acid leaching.

### 3.3. Effect of Roasting

After the pretreatment and nitric acid leaching process, roasting was performed at 200–700 °C for 10 h. The results of leaching from DI water are shown in Figure 7a. When roasting at 225 °C for 10 h, the lithium leaching ratio was the highest at 89.14%, and when roasting above 300 °C, the leaching ratio decreased. However, when roasting at 225 °C, both nickel and cobalt leached out, which is believed to be because some of the nickel nitrate and cobalt nitrate were not converted into metal oxides. Therefore, roasting at 275 °C is appropriate for selective lithium leaching.

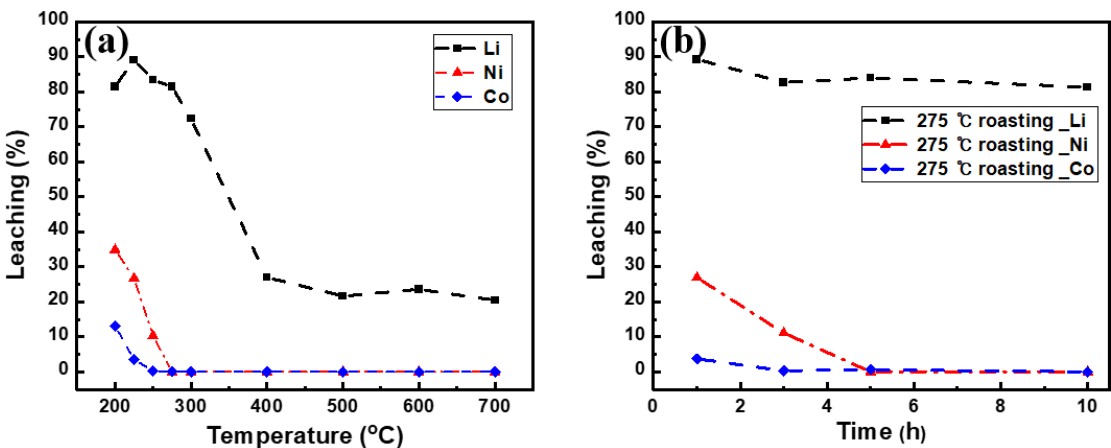

**Figure 7.** Variation in leaching ratio with (**a**) roasting temperature and (**b**) roasting time at 275 °C.

The nitrate compound is converted into a metal oxide at a specific temperature during roasting after nitric acid leaching according to the below reaction. It can then be decomposed into oxides that are not leached in DI water through roasting.

$$M(NO_3)_2(s) \rightarrow MO(S) + 2NO_2(g) + 1/2\,O_2(g) \tag{3}$$

In the case of LiNO$_3$, the temperature required for decomposition into oxides is high; thus, only Li can be leached when roasting is performed at certain temperatures.

For TG-DSC analysis, black powder was leached in 5 M nitric acid, and then dried at 100 °C for 10 h. As a result of the analysis, it can be seen that most nitric acid compounds undergo endothermic reactions at temperatures below 300 °C. To determine the temperature at which each nitrate compound is converted to a metal oxide, TG-DSC analysis was performed; the results are shown in Figure 8. In the case of LiNO$_3$, the temperature exceeds 600 °C, and other nitrate compounds such as Ni(NO$_3$)$_2$, Co(NO$_3$)$_2$, and Mn(NO$_3$)$_2$

decompose at temperatures below 300 °C [12]. Decomposed metals exist as oxides that cannot be leached in deionized (DI) water. To confirm the conversion of lithium nitrate, the black powder was leached in nitric acid and roasted at 200–700 °C. The results of the XRD analysis are shown in Figure 9. The results confirm that the lithium nitrate peak did not appear at temperatures above 400 °C. The significant decrease in the lithium leaching ratio owing to the roasting process above 400 °C was investigated. $LiNO_3$ and nitrate compounds, $Ni(NO_3)_2 \cdot 6H_2O$, $Co(NO_3)_2 \cdot 4H_2O$, and $Mn(NO_3)_2 \cdot 4H_2O$ were mixed and roasted at 400 °C, followed by leaching in DI water. The results of the XRD analysis of the undissolved residue are shown in Figure 10. Nitrate compounds other than manganese nitrate decomposed during roasting, and oxide peaks, such as NiO and $Co_3O_4$, appeared. In the case of manganese nitrate, it was confirmed that $LiMn_2O_4$ was formed by reaction with lithium nitrate when roasting at temperatures above 400 °C. Based on this analysis, it was confirmed that the formation of $LiMn_2O_4$ led to a decrease in the lithium leaching ratio. Therefore, roasting at 275 °C is suitable.

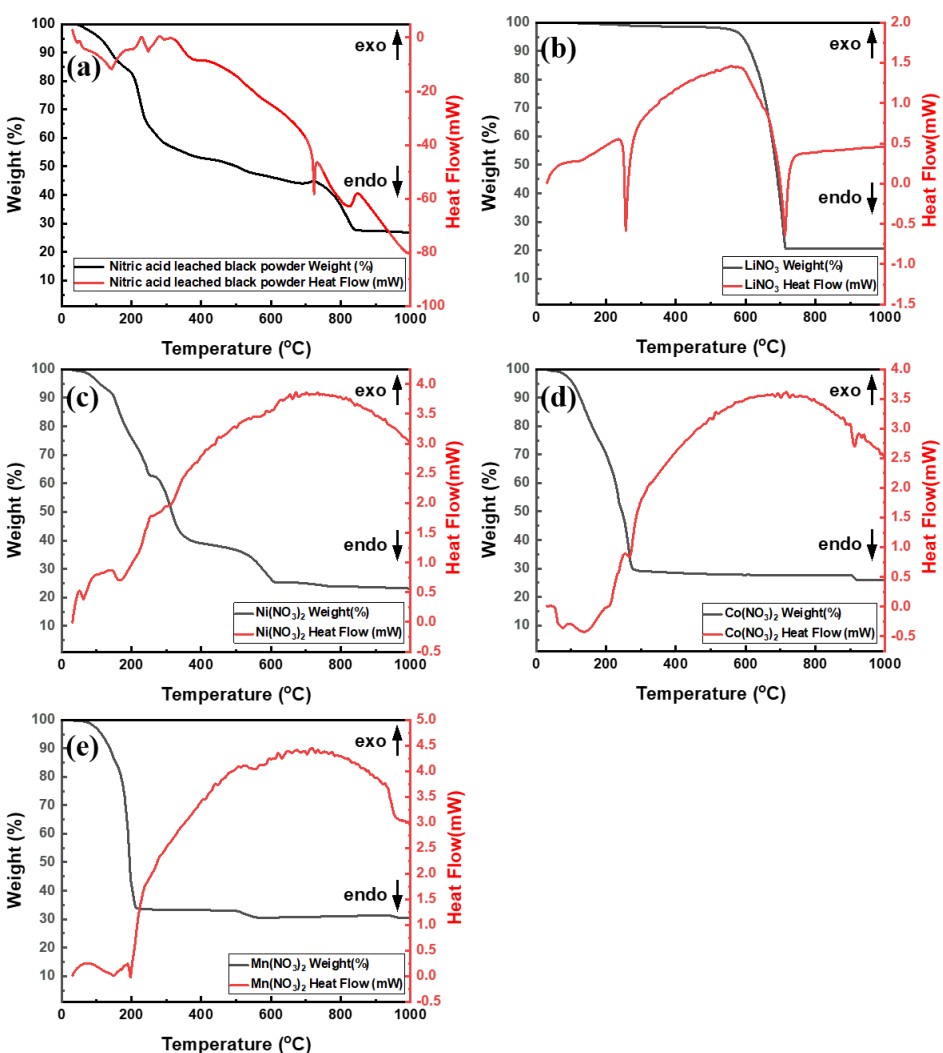

**Figure 8.** Thermogravimetric analysis of black powder ((**a**) 5 M nitric acid leached black powder) and nitrate compounds ((**b**) $LiNO_3$, (**c**) $Ni(NO_3)_2$, (**d**) $Co(NO_3)_2$, (**e**) $Mn(NO_3)_2$).

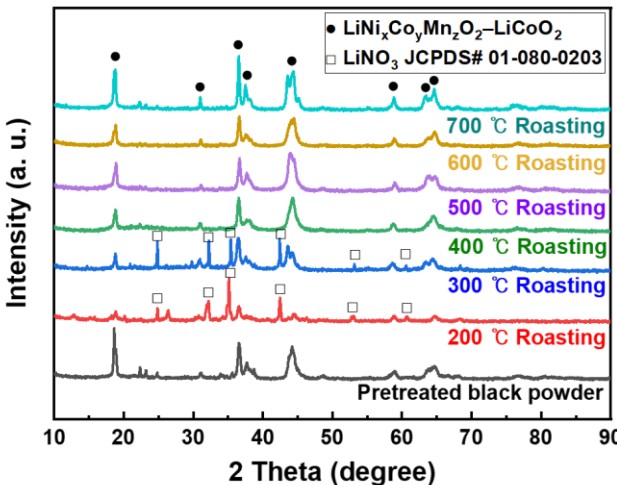

**Figure 9.** XRD for various roasting temperature values.

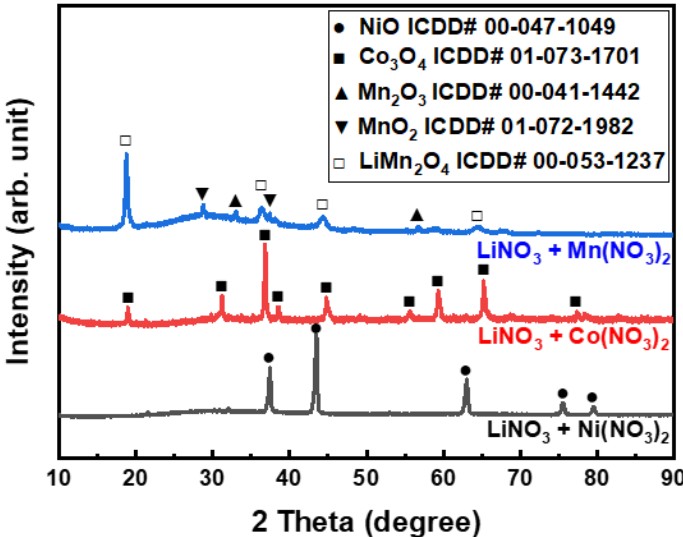

**Figure 10.** XRD results of leaching residue after roasting at 400 °C for lithium nitrate and nitrate compound.

When roasting at 275 °C, the effect of varying the leaching ratio of metal with time is shown in Figure 7b. Roasting for more than 10 h was suitable for selective lithium leaching, and the lithium leaching rate was 80.46%. When roasting at 225 °C, the leaching rate is approximately 9% lower; however, this approach can prevent nickel and cobalt leaching.

### 3.4. Effect of Solid–Liquid Ratio

After roasting, leaching in DI water at room temperature at ratios of 1–30 mL/g is shown in Figure 11. In theory, the solubility of $LiNO_3$ in DI water is 900 g/L, which is very high compared to that of $Li_2CO_3$ (13 g/L). Therefore, when 3.92% of lithium in black powder is converted to $LiNO_3$, leaching is possible with 0.4 mL/g of DI water. However, the use of a very small amount of DI water makes it difficult to stir, and results in the problem whereby the powder and the solution do not come into contact. In this experiment, it was determined that the use of 10 mL/g of DI water was suitable, and it showed a lithium leaching rate of 80% or more. Lithium nitrate leaching through this process has a difference of more than two times the leaching rate compared to the lithium carbonate process [13] using the carbon reduction reaction. Therefore, selective lithium leaching can be efficiently performed through a selective lithium nitrate recovery process.

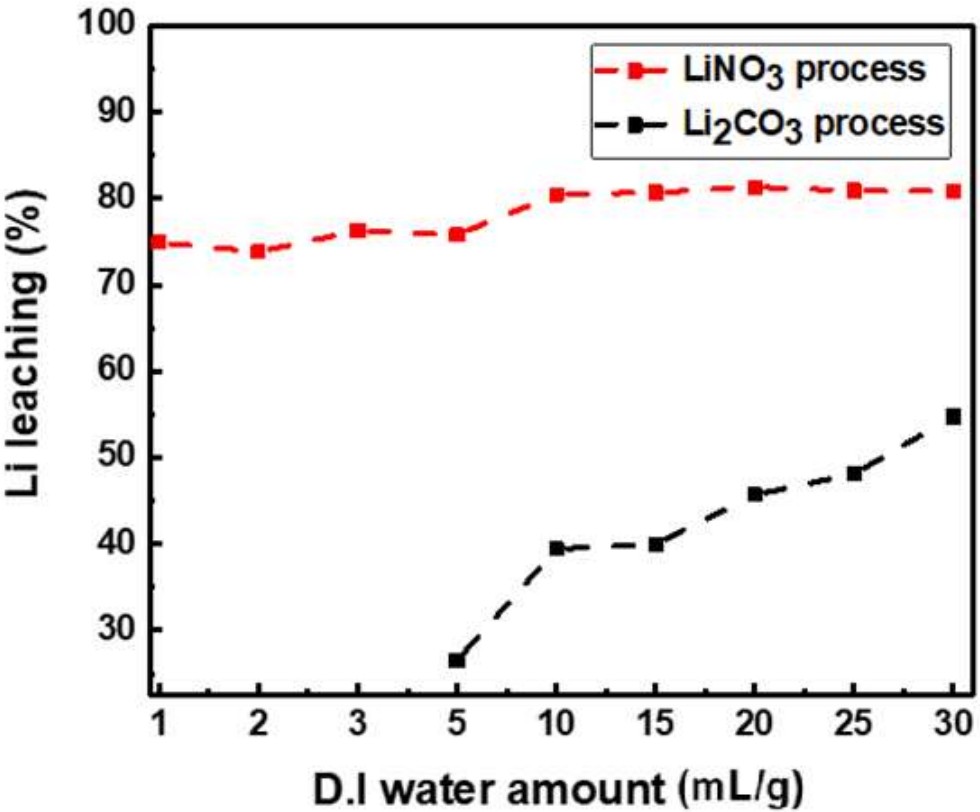

**Figure 11.** Variation in selective lithium leaching ratio with lithium nitrate leaching process and lithium carbonate leaching process [13].

*3.5. Recovered Powder Analysis*

Selective lithium leaching took place at 600 °C for 10 h pretreatment, 10 M nitric acid leaching (2 mL/g), roasting for 10 h at 275 °C, and leaching in DI water (10 mL/g). From the results of the ICP analysis after solid–liquid separation, 80.46% of lithium leaching occurred, and no other metal leaching occurred. Other metals such as Ni, Co, Mn, and Al were not leached. The powder was obtained by drying the obtained liquid for 24 h at 90 °C. Lithium nitrate was observed in the XRD analysis, as shown in Figure 12. The results of ion chromatography (IC, 930 Compact IC Flex, Metrohm, Herisau, Swizerland) of the recovered powder are shown in Table 2. It was confirmed that more than 97% of the $LiNO_3$ powder was recovered. It was confirmed that trace amounts of Na, F, and $SO_4$ were present as impurities. Therefore, further studies should be conducted on the removal of residual impurities from the solution and powder to recover high-purity $LiNO_3$.

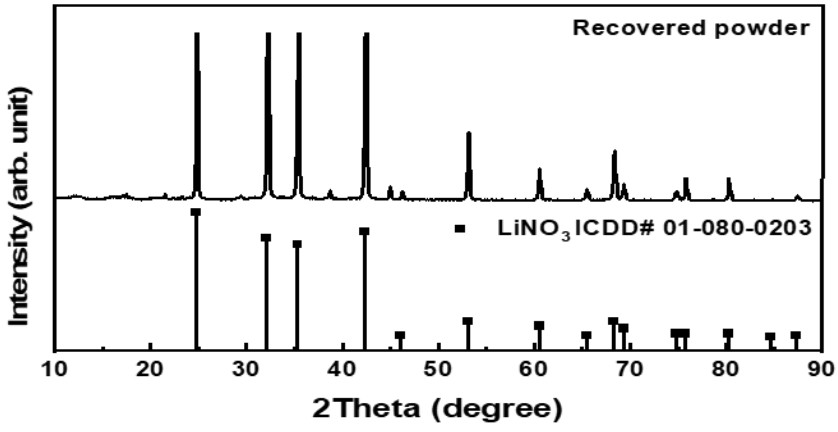

**Figure 12.** Powder XRD results obtained from solid–liquid separated solution.

**Table 2.** IC analysis result for recovered powder.

| IC Analysis | Component (mg/kg) | | | | |
|---|---|---|---|---|---|
| | Li | NO$_3$ | F | SO$_4$ | Na |
| Recovered LiNO$_3$ | 98,009 | 864,826 | 1982 | 2530 | 2472 |

## 4. Conclusions

The purpose of this study was to examine the selective lithium nitrate recovery process for efficient lithium reutilization from spent lithium batteries from different industries. The following conclusions were drawn.

1.  The organic matter in the black power led to a decrease in the lithium leaching rate. Thus, organic substances, including carbon, should be removed through pretreatment. To obtain optimal results, pretreatment should be conducted for more than 5 h at 600 °C.
2.  LiNi$_x$Co$_y$Mn$_z$O$_2$ and LiCoO$_2$ in the black powder were converted into nitrate compounds such as LiNO$_3$, Co(NO$_3$)$_3$, Ni(NO$_3$)$_2$, and Mn(NO$_3$)$_2$ by nitric acid leaching with more than 1 mL/g of 10 M nitric acid. The residual nitric acid that was not involved in the reaction was removed during the roasting process.
3.  With the exception of LiNO$_3$, the nitrate compounds were converted into oxides through roasting, allowing selective lithium leaching. Roasting at 275 °C for a minimum of 5 h was found to be adequate for selective lithium leaching. When leaching was performed after roasting above 400 °C, the leaching rate decreased significantly owing to the formation of LiMn$_2$O$_4$, which could not be leached in DI water.
4.  When the sample was leached with DI water (10 mL/g) after roasting, over 80% of the lithium was leached. This indicates that more than twice the amount of lithium can be recovered compared with the lithium carbonate recovery method by employing carbon reduction at the same solid–liquid ratio.

The leached liquid recovered through selective lithium leaching was dried at 90 °C for 24 h to obtain a white powder. The powder was analyzed and identified as LiNO$_3$ with a purity of 97%.

**Author Contributions:** Conceptualization, Y.J., B.Y., S.P., Y.K. and S.S.; Data curation, Y.J.; Formal analysis, Y.J.; Investigation, B.Y., S.P. and Y.K.; Methodology, Y.J.; Project administration, Y.K. and S.S.; Supervision, Y.K. and S.S.; Validation, B.Y., S.P., Y.K. and S.S.; Visualization, S.P. and Y.K.; Writing—original draft, Y.J.; Writing—review and editing, Y.K. and S.S. All authors have read and agreed to the published version of the manuscript.

**Funding:** This study was supported by the Technology Innovation Program (Development of Material Component Technology) (Project No. 2011183) funded by the Ministry of Trade, Industry, and Energy, Republic of Korea.

**Institutional Review Board Statement:** Not applicable.

**Informed Consent Statement:** Not applicable.

**Data Availability Statement:** Not applicable.

**Acknowledgments:** The authors are grateful for the support provided by the Ministry of Trade, Industry, and Energy, Republic of Korea.

**Conflicts of Interest:** The authors declare no conflict of interest.

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
