# Peer review of "Study on Roasting for Selective Lithium Leaching of Cathode Active Materials from Spent Lithium-Ion Batteries"

_metals, doi:10.3390/met11091336_

Round 1
Reviewer 1 Report
The manuscript was revised by the authors and submitted for re-review. The authors' comments indicate that the changes made relate mainly to language details and formatting. Content adjustments and additions (ion chromatography analysis) were also made.
The revised manuscript includes new data for the chemical composition of the NMC under study. On page 2 line 7 the authors report "trace amounts of Al,Fe, and Cu". ICP-OES is capable to determine trace amounts very precisely, however, a value for Cu is not given. But 3.20% Al is described as "trace amount".
In the manuscript, it is probably not intended to give a stoiciometry of the investigated NMC according to ICP-OES analysis. A calculation gives an approximate stoichiometry of Li1.03Ni0.49Mn0.25Co0.26. What the analysis does not explain and is not further addressed in the manuscript is the high mass fraction of about 43%, which is not explained by the presence of the NMC, the Al and the Fe. The assignment of the reflections in Fig. 1 remains equally vague; the formula "LiNixCoyNizO2 - LiCoO2" given there does not allow any indication of the identification of the NMC found. In contrast, diffractograms can be found in the literature which explain the measured pattern with a high degree of certainty.
Figure 5 was taken from the earlier manuscript without any changes, only the earlier indexing of the reflections was removed. The diagram confirms that carbon was successfully removed, however, what happens to the NMC? The peaks appear broader and less intense?
Page 7 line 167.169: This sentence makes no sense. Where comes the LiCoO2 from, it seems not to be present in the material?
Lines 171/172: What is the role of silver nitrate? AgNO3 is even not mentioned in the experimental section. Please explain this statement.
Figure 6 shows the amount of lithium obtained after treatment with different volumes of 10M HNO3. Adding such a high concentration of HNO3 to NMC will not only lead to the release of Li, but also to the partial dissolution of all NMC. Thus not only Li should go into solution, but also Ni, Mn and Co.
So, is it eally observed that Li is SELECTIVELY dissolved. What about Ni, Co, Mn?
Page 9: Black mass treated with HNO3 converts the NMC constituents into nitrate salts. Below 300°C the nitrates of Ni, Mn and Co are decomposed into oxides which ar less soluble in i.e. water. Only LiNO3 remains undecomposed and is easily soluable in water. But why is LiNO3 not observed in the treated and dried black mass in Fig. 9? Why does it need 300°C that it appears in the XRD? Why do not the nitrates Ni, Co and Mn appear in the diffraction pattern since they are present in much higher amounts according to Table 1?
Page 13 line 238: The given Li content does not correspond to table 1.
Why is Li2CO3 considered in section 3.4? It does not appear anywhere else in the manuscript?
Page 15, line 255: „Other metals such as Ni, Co, Mn, and Al were not leached.“ This seems to cotradict the results in Fig. 7a The material was treated under similar conditions but shows a solubility for Ni and probably Co.
In my opinion, the revision did not improve the quality of the manuscript enough to be published. I propose to reject this manuscript.
Author Response
Dear Reviewer,
Thank you for your useful comments and suggestions on the language and structure of our manuscript. We have modified the manuscript accordingly, and detailed corrections are listed below point by point:
Point 1: The revised manuscript includes new data for the chemical composition of the NMC under study. On page 2 line 7 the authors report "trace amounts of Al,Fe, and Cu". ICP-OES is capable to determine trace amounts very precisely, however, a value for Cu is not given. But 3.20% Al is described as "trace amount".
Response 1: The Cu content is 1.15 wt.%, and Copper content information is added in Table 1.
Point 2: In the manuscript, it is probably not intended to give a stoiciometry of the investigated NMC according to ICP-OES analysis. A calculation gives an approximate stoichiometry of Li1.03Ni0.49Mn0.25Co0.26. What the analysis does not explain and is not further addressed in the manuscript is the high mass fraction of about 43%, which is not explained by the presence of the NMC, the Al and the Fe. The assignment of the reflections in Fig. 1 remains equally vague; the formula "LiNixCoyNizO2 - LiCoO2" given there does not allow any indication of the identification of the NMC found. In contrast, diffractograms can be found in the literature which explain the measured pattern with a high degree of certainty.
Response 2: Because it uses the powder of the spent lithium ion battery, it was judged that various types of NCM powder were included, so it was labeled as LiNixCoyNizO2-LiCoO2. As a result of XRD peak analysis for black powder, the substances shown in the table below were found. In the case of black powder, it is difficult to indicate the exact XRD peak because various types of cathode active materials may be included.
|
Reference code (ICDD number) |
Chemical formula |
|
01-085-1980 |
Li0.75Ni1.05O2 |
|
01-088-1750 |
LiMnO2 |
|
01-075-0539 |
LiCoO2 |
|
98-016-0966 |
Li0.99 Ni0.46Mn1.541O4 |
|
01-085-1981 |
Li0.65Ni1.05O2 |
|
00-052-0457 |
Li1.15(MnxNi1-x)0.85O2 |
|
98-016-0966 |
Li0.99Mn1.541Ni0.46O4 |
|
01-070-4314 |
Li1.03Co0.1Ni0.77Mn0.1O2 |
Point 3: Figure 5 was taken from the earlier manuscript without any changes, only the earlier indexing of the reflections was removed. The diagram confirms that carbon was successfully removed, however, what happens to the NMC? The peaks appear broader and less intense?
Response 3: The peaks appeared broader and less intense. This is thought to be due to the formation of various types of lithium oxides through carbon reduction and reoxidation of oxides containing carbon and lithium. In the case of carbon, the peak disappeared because it was removed with CO and CO2.
Point 4: Page 7 line 167.169: This sentence makes no sense. Where comes the LiCoO2 from, it seems not to be present in the material?
Response 4: Corrected the sentence. LiCoO2 is actually a waste powder, so it contains various cathode active materials such as NCM, LCO, and LMO. Prior to this experiment, LCO and NCM reagents were also tested, and there was no problem in recovering LiNO3.
Point 5: Lines 171/172: What is the role of silver nitrate? AgNO3 is even not mentioned in the experimental section. Please explain this statement.
Response 5: Checked and corrected.
Point 6: Figure 6 shows the amount of lithium obtained after treatment with different volumes of 10M HNO3. Adding such a high concentration of HNO3 to NMC will not only lead to the release of Li, but also to the partial dissolution of all NMC. Thus not only Li should go into solution, but also Ni, Mn and Co.
Response 6: Figure 6 shows the results after selective lithium leaching including roasting. Ni, Mn, and Co were not leached in D.I water. Roasting is the result of 10 hours of roasting at 275 oC and leaching with 10 ml/g of D.I water.
Point 7: Page 9: Black mass treated with HNO3 converts the NMC constituents into nitrate salts. Below 300°C the nitrates of Ni, Mn and Co are decomposed into oxides which ar less soluble in i.e. water. Only LiNO3 remains undecomposed and is easily soluable in water. But why is LiNO3 not observed in the treated and dried black mass in Fig. 9? Why does it need 300°C that it appears in the XRD? Why do not the nitrates Ni, Co and Mn appear in the diffraction pattern since they are present in much higher amounts according to Table 1?
Response 7: The black powder shown in Figure 9 is the powder before leaching into 10M nitric acid. Therefore, the LiNO3 peak does not appear. And XRD analysis is not possible immediately after leaching in 10M nitric acid, and analysis is possible after it is changed to a solid through constant roasting. This allowed XRD analysis after roasting above 200 oC. When roasted at 200 oC, some conversion to oxide occurred, and it was confirmed that it exists in the form of nitrate. It was confirmed that the leaching rate of Ni and Co was reduced when roasting at 200 - 250 oC for 10 - 50 hours, which can be seen in Figure 8 in the article.
Point 8: Page 13 line 238: The given Li content does not correspond to table 1.
Response 8: Checked and corrected.
Point 9: Why is Li2CO3 considered in section 3.4? It does not appear anywhere else in the manuscript?.
Response 9: In the case of Li2CO3, the solubility is very low at 13 g/L and the recovery rate of lithium carbonate is low, but in the case of LiNO3, the solubility is very high at 900 g/L and the recovery rate of lithium nitrate is high. In other words, in the lithium recovery method through this study, efficient leaching is possible at a low leaching high-liquid ratio of D.I water, and it was decided that an explanation is necessary.
Point 10: Page 15, line 255: „Other metals such as Ni, Co, Mn, and Al were not leached.“ This seems to cotradict the results in Fig. 7a The material was treated under similar conditions but shows a solubility for Ni and probably Co.
Response 10: After roasting at 275 oC for 10 h, only lithium was leached when leaching with D.I water, and leaching of other metals did not occur. Figure 7 shows the same result.
Reviewer 2 Report
After corrections were introduced I recommend to accept the manuscript to publication in Metals (MDPI), although I still found two minor errors:
- Chemical formula "LiNixCoyNizO2" in Figure 5 is incorrect.
- "exo/endo" convention should be denoted not only in Fig. 3, but also in Fig. 8.
Author Response
Dear Reviewer,
Thank you for your useful comments and suggestions on the language and structure of our manuscript. We have modified the manuscript accordingly, and detailed corrections are listed below point by point:
Point 1: Chemical formula "LiNixCoyNizO2" in Figure 5 is incorrect.
Response 1: correction has been completed.
Point 2: "exo/endo" convention should be denoted not only in Fig. 3, but also in Fig. 8.
Response 2: Relevant information is reflected.
Reviewer 3 Report
correction has done and paper quality improved. it is fine to be published.
Author Response
Dear Reviewer,
Thank you for your useful comments.
Reviewer 4 Report
This is an interesting paper on the development of an efficient method for lithium recovery from exhausted batteries.
The experimental part is sufficiently described and the English language is of a good level. Conclusions are well supported by experimental data.
I have only one comment: lithium metal oxides are not the only type of cathode used in commercial lithium batteries. Indeed, LiFePO4 and, even more, difficult to be recycled, lithium-multimetal-phosphate cathodes are currently used and developed (see for example Journal of the Electrochemical Society 144 (1997) 1188-1194 DOI:10.1149/1.1837571, J. Mater. Chem. A 8 (2020) 25727-25738 DOI:10.1039/D0TA06865A, Journal of Cleaner Production 316 (2021) 128098 DOI:10.1016/j.jclepro.2021.128098, Adv. Funct. Mater. 25 (2015) 4032-4037 DOI:10.1002/adfm.201501167, ACS Sustainable Chemistry and Engineering 9 (2021) 4711-47215 DOI:10.1021/acssuschemeng.0c08487, Green Chemistry 23 (2021) 1344-13527 DOI:10.1039/d0gc03683h, Electrochim. Acta 225 (2017) 533-542 DOI:10.1016/j.electacta.2016.12.149).
The authors must comment on the utilization of the proposed approach on this type of cathode materials and if it is expected to observe any difference.
Author Response
Dear Reviewer,
Thank you for your useful comments and suggestions on the language and structure of our manuscript. We have modified the manuscript accordingly, and detailed corrections are listed below point by point:
Point 1: I have only one comment: lithium metal oxides are not the only type of cathode used in commercial lithium batteries. Indeed, LiFePO4 and, even more, difficult to be recycled, lithium-multimetal-phosphate cathodes are currently used and developed (see for example Journal of the Electrochemical Society 144 (1997) 1188-1194 DOI:10.1149/1.1837571, J. Mater. Chem. A 8 (2020) 25727-25738 DOI:10.1039/D0TA06865A, Journal of Cleaner Production 316 (2021) 128098 DOI:10.1016/j.jclepro.2021.128098, Adv. Funct. Mater. 25 (2015) 4032-4037 DOI:10.1002/adfm.201501167, ACS Sustainable Chemistry and Engineering 9 (2021) 4711-47215 DOI:10.1021/acssuschemeng.0c08487, Green Chemistry 23 (2021) 1344-13527 DOI:10.1039/d0gc03683h, Electrochim. Acta 225 (2017) 533-542 DOI:10.1016/j.electacta.2016.12.149).
The authors must comment on the utilization of the proposed approach on this type of cathode materials and if it is expected to observe any difference.
Response 1: We are planning to research on various powders such as LiFePO4 and lithium mulltimetal-phosphate cathodes. In the future, as various powders are secured, we are planning to study the leaching results and control methods for impurities.
This manuscript is a resubmission of an earlier submission. The following is a list of the peer review reports and author responses from that submission.
Round 1
Reviewer 1 Report
Dear authors,
this paper shows results of recovering lithium with using nitrate and this topic can be important.
this paper can be published after following correction.
Figure 4, Time (H) -> Time (h)
Figure 6 Amount of nitrate acid -> Amount of nitric acid?
Figure 7 TIME (Hour) -> Time (h)?
Figure 9 and 10, graph font is different.
References,
chemical formula, such as Li2O3, 2 and 3 should be subscript.
references 9,10,11, author name is full name, but other part shows initial and family name. it should be consistent.
references 13, & is not necessary.
Reviewer 2 Report
Manuscript „A study on nitration roasting for selective lithium leaching of cathode active materials from spent lithium-ion batteries” by YJ Jung et al. reports lithium recovery process from spent Li-ion cells. Recently recycling of spent lithium batteries became urgent issue so I have no doubts about the significance of the topic. Having said this, I would like to point out that the results are not clearly presented and so major correction is required. Some specific points are listed below:
The whole manuscript should be thoroughly checked in terms of English language and style of writing to make it more appealing to the readers and correct the errors, e.g. Page 7 Line 178 “However, after the black powder, there was sedimentation with nitric acid(…)”.
Page 2, Line 55. Description “Spent LIB powder (black powder)” is not sufficient to characterize the material investigated in this work. To make the presented results interesting for broader audience, the authors are requested to provide more information: method of recovery and separation form the cell, is it only cathode or mixed cathode and anode materials, other treatment procedures, composition: binder, residual electrolyte, Al foil, was it from single type of a battery, or mixture from different cells.
Page 2, Line 59. Phrase “the phase of the powder” should be replaced by more adequate “composition of the powder”.
Page 2, Lines 62-63 and Figure 1. In the text authors suggest detection of NMC, i.e. LiNixMnyCozO2 oxide, whereas in Fig. 1 peaks are identified and carbon and Co0.2069Li0.756Ni0.793O2 (no Mn present) – this should be clarified. Location of peaks from the LiNi0.5Mn1.45O4 spinel should be better visualized, as they can be hardly seen in the graph.
Fig. 1. Reference pattern numbers from ICDD or other well recognized source should be provided in order to support analysis of the XRD pattern.
Page 3, Lines 92-98. Some important details of the sedimentation step are missing: was the suspension stirred or still and what was the sedimentation time.
Page 3, Lines 93-94. Composition of the “black powder” given in this section is inconsistent with previous description and with analysis presented in Fig. 1.
DSC curves (Figs. 3 & 8). Direction up or down of exothermic and endothermic effects should be noted in the DSC plots to avoid possible confusion, as different standards can be used in different laboratories
Fig. 5. Reference pattern numbers from ICDD or other well recognized source should be provided in order to support analysis of the XRD pattern.
Page 6, Line162. According to results presented in Fig. 7 leaching ratio decreased above 300°C, not above 400°C, as stated in this paragraph.
Fig. 9. Description of the detected components should be specific and unambiguous, with reference pattern numbers. Term “Metal oxide” is far from being clear.
Fig. 10. Term “Metal oxide (NiO, Co3O4, MnO2, Mn2O3)” should be replaced with specific marks for each of them, so that the reader could confirm the analysis.
Page 8, Line 203. Abbreviation “S:L” should be explained.
Page 9, Line 219. Term “Materialization” should be explained.
Page 9, Line 223. Amount of other metals (Ni, Co, Mn) in the resulting solution from ICP analysis should be given.
Page 9, Line 225. Abbreviation “IC” should be explained.
Reviewer 3 Report
I am sorry, but I cannot recommend the present manuscript for publication in METALS. Although the manuscript describes an interesting approach, it is characterized by considerable inconsistency in the presentation of results and by missing information or a disorganized presentation of information. It is recommended that the authors not resubmit the manuscript until it has been thoroughly revised.
- The most significant criticism is the inconsistency of the data presented.
Table 1 presents the detected elemental contents of the used black mass sample. It must be assumed that this composition is representative of the material that was studied in the present work. The statement (page 2, line 65) "The main metals identified in powder were Co, Ni, Mn, and Li along 65 with trace amounts of Al, Fe, and Cu." is incorrect. Co is the main component and all other elements can be called minor components. With contents of 3.34% for Al and 8.23% for Fe, these cannot be called trace components. Copper, mentioned in the text, is not found in the table.
It is important to clarify from which sources the detected elements originate. Does Al come from the shredding of the cathode foils, as is usual in the extraction of the black mass? Where does the high iron content come from? Such contents are typical for cathode materials to which LiFePO4 has been added. If so, it should be visible in the diffractogram in Fig. 1. But diffractogram is just too small to see the reflexes. Please refer to other publications, the reflexes should be at least 40% of the total height of the diagram. Such a representation is absolutely insufficient.
The assignment of the reflexes in Fig. 1 is not understandable. The reflexes are explained with two components and with the formulas "Co0.2069Li0.756Ni0.793O2" and "LiNi0.5Mn1.5O4". In contrast, the analysis in Table 1 shows only a low Ni content of 2.07% and an even lower Mn content of 1.6%. Even though the proportions of these components were not quantified, the molar ratios of the elements cannot explain these claimed identified compounds.
Fig. 5 is equally inconsistent. Again, the diffractograms are much too small for the reflections to be discernible. Is it plausible to assume that sufficient quantities of Li2CO3 were formed after treatment at 600°C? Is it plausible to identify Li2CO3 by only one reflex? What gas atmosphere was chosen for the treatment? Likewise, "LiAlO2" is identified based on only one reflex, moreover, the Al content is quite low and it is unlikely that it would form LiAlO2 from an Al content of 3.34% at these temperatures. Or was it already present before? It is not clear from Fig. 1. Finally, it is completely unclear which compound is supposed to be hidden behind the assignment "LiNixMnyCozO2 - LiCoO2". In any case, give the JCPDS number for each compound for identification!
The element contents in Table 1 add up to 52.67 wt% without copper. Since Ni, Mn, and Co are present as oxides, the percentage of inorganic constituents will be somewhat higher. If Ni, Mn and Co are assumed to be NiO, MnO and CoO, the resulting content is about 75wt%. The rest is binder and carbon. How then can thermal treatment at 800°C result in a mass loss of 44.76% (Fig. 3 and page 3, line 127)?
Fig. 9 shows the dirffaction pattern of "pretreated black powder" obtained after HNO3 treatment and used for roasting. Why was this diffractogram not analyzed? Which compounds are present? And then the diffractogram shows the reflections for LiNO3, where Li is present only with 4.52% of the starting material. Is there no evidence of compounds of Co, Ni, Mn, Fe? And then Ni, Co, Mn show up again at 700°C undefined as "LiNixCoyMnzO2", also without giving more details about this compound. In the literature, there are many powder diffractograms of NMC materials with different proportions of Ni, Mn and Co.
Fig. 8 shows TG curves of the thermal decomposition of nitrates of Li, Ni, Mn, Co. No information has been given on the measurement conditions, and it is also unclear whether the salts used contain water of crystallization or not.Why were the TG curves of the treated black mass not shown?
- Incomplete data and designations
TG measurements have a significant part in the manuscript. Neither the instrument used nor the measurement parameters were mentioned, such as heating rate and gas atmosphere.
Chapter "2.2 Experimental Procedure" is difficult to understand. The individual steps are discussed, but there is no isolated and clear description. "2.2.1 Pretreatment" contains statements about "sedimentation" and "roasting", "22.2 Sedimantation" contains statements about "roasting".
All statements must be written in such a way that each individual work step can be reproduced by an independent working group.
Why is the term "sedimentation" used? The reaction of black mass with 10N HNO3 yields a suspension in the cold, which is more or less viscous due to the chosen ratio of mass and volume. Sedimentation does not take place in the true sense of the word.
Fig. 2: An IC analysis is not shown in the manuscript.